# Deaths and cardiopulmonary events following colorectal cancer screening—A systematic review with meta-analyses

Frederik Handberg Juul Martiny[1,2]*, Anne Katrine Lykke Bie[1], Christian Patrick Jauernik[1], Or Joseph Rahbek[1], Sigrid Brisson Nielsen[1], Emma Grundtvig Gram[1,3], Isabella Kindt[1], Volkert Siersma[1], Christine Winther Bang[1], John Brandt Brodersen[1,3,4]

1 Department of Public Health, Section of General Practice and Research Unit for General Practice, University of Copenhagen, Copenhagen, Denmark, 2 Department of Social Medicine, Bispebjerg and Frederiksberg Hospital, Copenhagen, Denmark, 3 Research Unit for General Practice in Region Zealand, Copenhagen, Denmark, 4 Department of Community Medicine, Faculty of Health Sciences, Research Unit for General Practice, UiT The Arctic University of Norway, Tromsø, Norway

☉ These authors contributed equally to this work.
* fhm@sund.ku.dk

**Data Availability Statement:** he data underlying the results presented in the study are available from an Open Science Framework project via https://osf.io/89zxq/.

## Abstract

### Background

Colorectal cancer screening programmes (CRCSPs) are implemented worldwide despite recent evidence indicating more physical harm occurring during CRCSPs than previously thought. Therefore, we aimed to review the evidence on physical harms associated with endoscopic diagnostic procedures during CRCSPs and, when possible, to quantify the risk of the most serious types of physical harm during CRCSPs, i.e. deaths and cardiopulmonary events (CPEs).

### Methods

Systematic review with descriptive statistics and random-effects meta-analyses of studies investigating physical harms following CRCSPs. We conducted a systematic search in the literature and assessed the risk of bias and the certainty of the evidence.

### Results

We included 134 studies for review, reporting findings from 151 unique populations when accounting for multiple screening interventions per study. Physical harm can be categorized into 17 types of harm. The evidence was very heterogeneous with inadequate measurement and reporting of harms. The risk of bias was serious or critical in 95% of assessments of deaths and CPEs, and the certainty of the evidence was very low in all analyses. The risk of death was assessed for 57 populations with large variation across studies. Meta-analyses indicated that 3 to 23 deaths occur during CRCSPs per 100,000 people screened. Cardio-pulmonary events were assessed for 55 populations. Despite our efforts to subcategorize CPEs into 17 distinct subtypes, 41% of CPE assessments were too poorly measured or

**Funding:** The main author (FHJM) received financial support via the research grant "Sara Krabbes legat" from the Danish Society for General Practitioners (https://www.dsam.dk/forskning/sara_krabbes_legat/), covering expenses related to Open Access publication. The Danish Cancer Society Research Center (https://www.cancer.dk/forskning/stoette-til-forskning/funding/) funded one year's salary for FHJM to conduct the systematic review, and the William Demant Foundation (https://www.williamdemantfonden.dk/) supported FHJM's participation in the Preventing Overdiagnosis Conference 2017 in Quebec, Canada. The funders had no role in study design, data collection and analysis, decision to publish, or preparation of the manuscript. The first author is independent of the funding bodies.

**Competing interests:** All authors have completed the ICMJE uniform disclosure form at www.icmje.org/coi_disclosure.pdf and declare: the first author had financial support from the Danish Society for General Practice, the Danish Cancer Society and the William Demant Foundation for the submitted work, there were no financial relationships with any organizations that might have an interest in the submitted work in the previous three years; no other relationships or activities that could appear to have influenced the submitted work.

reported to allow quantification. We found a tendency towards lower estimates of deaths and CPEs in studies with a critical risk of bias.

## Discussion

Deaths and CPEs during CRCSPs are rare, yet they do occur during CRCSPs. We believe that our findings are conservative due to the heterogeneity and low quality of the evidence. A standardized system for the measurement and reporting of the harms of screening is warranted.

## Trial registration

**PROSPERO Registration number** CRD42017058844.

## 1. Introduction

Colorectal cancer (CRC) is the third most commonly diagnosed cancer globally and the second most deadly [1]. Consequently, many countries have implemented colorectal cancer screening programmes (CRCSPs) to reduce mortality [2, 3]. Multiple screening strategies for colorectal cancer exist [4, 5]. Here, we focus on the most widely implemented types of CRCSPs: once-only Total Colonoscopy (TConly), once-only Flexible Sigmoidoscopy (FS), Total Colonoscopy following faecal occult blood testing (TCfobt) and Total Colonoscopy following sigmoidoscopy or other screening tests (TCfollowup).

Many international bodies recommend CRCSPs [3, 6–10] based on the evidence of their benefit, which has been compiled in numerous systematic reviews [4, 11–19]. Still, many people attending screening are not likely to benefit, and all risk being harmed unintentionally [20]. The evidence about screening tends to be skewed in favour of the benefits of screening, with little scientific attention to the harms of screening [21, 22]. Compared to the evidence about the benefits of screening, the evidence concerning harm is much less frequently investigated, often selectively and inadequately reported in studies and communicated in an unbalanced way in both randomised studies [23], guidelines [24] and invitation material for the public [25–28]. In addition, it has been highlighted in the PRISMA-harms extension to the reporting guideline for systematic reviews that systematic reviews tend to compound poor reporting of harm in primary studies, thus providing an inadequate account of harm associated with medical interventions [29].

In the case of CRCSPs, harm has been scarcely investigated, leading to uncertainty about the types of harm that may occur, the risk of these harms and their severity, collectively referred to as the extent of harm [11–19]. Thus, the most prevalent type of harm, i.e. physical harm, e.g. endoscopy-related complications [30], might be underreported. Consequently, the true extent of physical harm in the real-world setting may not be adequately reflected by the best-available evidence due to its known limitations noted above. Therefore, we conducted a systematic review following recommendations from the PRISMA harms extension that aimed to 1) identify all physical harms associated with CRCSPs via the scientific literature, 2) give an overview of key characteristics of the evidence about the harms of CRCSPs, 3) assess the quality of measurement and reporting of physical harms in studies, 4) determine whether characteristics of the screening intervention or the screening population affect the risk for physical harm or the consequences thereof, 5) evaluate the risk of bias and the certainty of the evidence

of findings from studies, 6) and 7) to quantify the risk and the consequences of physical harm when possible.

Due to an unexpectedly large heterogeneity of the evidence, we chose to divide the reporting of the review's findings into separate publications. This publication reports the overall findings of the review and findings related to the second to seventh objectives for the two most severe types of physical harm associated with CRCSPs, i.e. death and cardiopulmonary events (CPEs). Other findings will be reported in separate publications [31]. The design of the study was published online in abstract format in 2019 [32].

## 2. Methods

Conduct and reporting followed guidance from the Cochrane Handbook [33], the PRISMA-harms extension and the PRISMA 2020 reporting guideline [29, 34], the AMSTAR checklist [35] and scientific literature concerning the methodological challenges of reviewing the harms of interventions [22]. Before data extraction, we published a protocol for the systematic review on PROSPERO [36]. PROSPERO Registration number CRD42017058844.

### 2.1 Study eligibility

Studies eligible for review investigated any physical harm, e.g. bleeding or perforation of the bowel, occurring during or after the diagnostic screening procedures, i.e. sigmoidoscopy or colonoscopy, of people at average risk of colorectal cancer, i.e. a general screening population. Details about how screening and harms were operationalised and criteria for in- and exclusion are available in Appendix 1 in S1 File.

### 2.2 Information sources and search strategy

The search strategy was developed and assisted by an information specialist in the databases MEDLINE (via PubMed), Embase, CINAHL, PsycINFO and the Cochrane Library (Appendix 2 in S1 File). We searched all databases from their inception date to the 4th of March 2022 without restrictions concerning the publication date, language or study design [37]. All studies identified via the search strategy were compiled in Endnote, where duplicates were removed. The first author looked for studies missed by the search strategy in the reference list of studies included for review and in the reference list in former systematic reviews in the area. Studies deemed potentially eligible were subsequently assessed by a second reviewer. We did not look for unpublished studies or search for grey literature due to the magnitude of the evidence yielded by our systematic search.

### 2.3 Ongoing studies

We identified ongoing RCTs and systematic reviews relevant to the research question via included studies, the WHO ICTRP Search Portal, the PROSPERO register and our research network (Appendix 9 in S1 File).

### 2.4 Study selection

Two reviewers independently assessed all studies for eligibility, consulting a third reviewer in case of disagreement. We contacted study authors if full-text studies were unavailable. Studies published in languages other than Scandinavian or English were assessed for eligibility via colleagues speaking the language. We extracted the most recent data if multiple publications reported findings from the same study population. In case we could not retrieve full-text

articles, we included the abstracts and assessed these by the same methods as full-text articles. The reasons for excluding studies after full-text reading are listed in Appendix 3 in S1 File.

## 2.5 Data collection process

We used the PRISMA-harms extension [29] and a generic data collection template from the Cochrane Collaboration [38] to develop the data collection process (Appendix 4 in S1 File). We pilot-tested the data collection template on five of the included studies and subsequently adapted it to increase inter-reviewer reliability and validity, i.e. extracting data of interest to the review's aim. Study characteristics were extracted by the first author and verified by a second author. Two authors independently extracted all outcome data.

## 2.6 Risk of bias

We assessed the risk of bias on the outcome level using an internal guidance document developed from the ROBINS-I tool, developed by the Cochrane Collaboration, for risk of bias assessment of findings from non-randomised studies [39, 40]. Most studies on CRCSPs did not assess harm in the unscreened group. Therefore, we used the ROBINS-I tool for both randomised and non-randomised studies (NRSs) and excluded the bias domains about confounding. Two authors independently assessed the risk of bias for each of the six bias domains: Inception bias, misclassification bias, performance bias, missing data bias, measurement bias, and selective reporting bias. Of note, publication bias, i.e. the selective publication of research studies based on their results, is assessed via the GRADE approach (Appendix 5 in S1 File). The risk of bias was categorised as either low, medium, serious or critical, and the expected direction of the effect that bias might have on the estimate of harm was: overestimated, underestimated or unpredictable.

## 2.7 The certainty of evidence across studies included for review

We used the GRADE approach [41] to assess the certainty of evidence across included studies. We did not assess the certainty of evidence of findings from studies with a critical risk of bias in line with the ROBINS-I recommendation [39]. We could not calculate a baseline occurrence of death and CPEs due to the various countries, regions and times of conduct for the included studies. Also, because most studies did not assess harm in the unscreened group, we had to modify the design of the standard evidence profile tables, excluding the following items: 1) Baseline risk, 2) Relative risk, 3) Absolute risk reduction, and 4) Study design.

In line with the GRADE handbook, all included studies started with a GRADE rating of "low certainty" due to being one-armed, effectively providing evidence comparable to observational studies [41]. We also present the sum of all the downgrading factors as an indicator of the certainty of the evidence in addition to the usual four evidence grades, e.g. high, moderate, low, and very low (Appendix 5 in S1 File).

## 2.8 Categorisation of extracted data from studies

**2.8.1. Distinct screening interventions.** Following data extraction, we arranged studies into groups according to the screening interventions provided to people (Fig 1).

We created four groups: sigmoidoscopy (FS), once-only total colonoscopy (TConly), colonoscopy following an abnormal faecal occult blood test (TCfobt) and a colonoscopy after FS or other types of screening tests, e.g. barium enema (TCfollowup).

**2.8.2. Subpopulations in studies.** In some studies, part of the study population received one screening intervention, whereas the other part received another screening intervention,

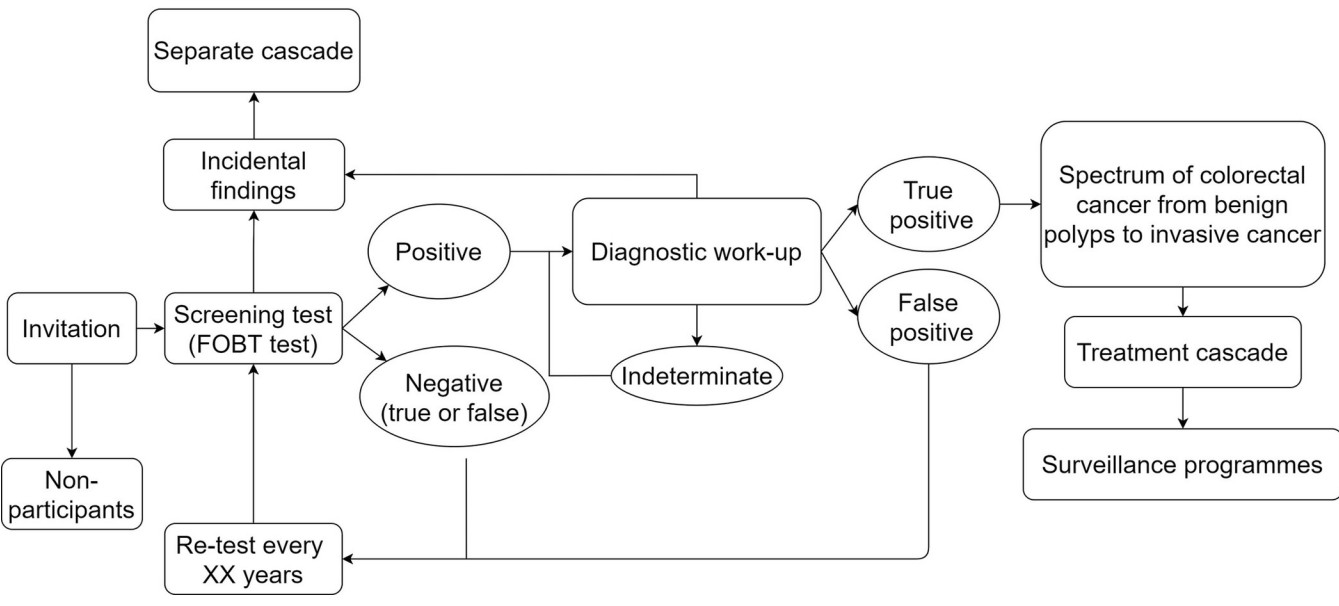

**Fig 1. CRC screening cascade.**

e.g. some received sigmoidoscopy, whereas others received a once-only colonoscopy as an alternative procedure. Therefore, we use the term "subpopulation" to refer to each study population receiving a screening intervention. In effect, there are more subpopulations than studies.

**2.8.3. Categorising and subcategorising physical harms.** First, we extracted data about the various types of physical harm due to CRCSPs. We categorised these into seven types of harm a priori: Death, Cardiopulmonary events, Perforation, Bleeding, Pain, Discomfort and Other types of harm. Next, two reviewers independently scrutinised all types of harm categorised in the category "Other", and if three or more studies reported on the same type of harm, we created a new harm category. Subsequently, each harm category, e.g. cardiopulmonary events (CPEs), was further subcategorised into subtypes of harm, e.g. vasovagal reactions, stroke, etc. (Appendix 6 in S1 File). We used existing guidelines about adverse events associated with colonoscopy [42–44] to categorise and subcategorise harm categories.

**2.8.4. People and procedures.** We used studies that reported the number of procedures performed and the number of people screened to calculate the mean number of procedures per person (Appendix 7 in S1 File). Subsequently, we used the mean number of procedures per person to estimate the number of people screened in studies that only reported the number of procedures provided.

**2.8.5. Correlation and causality.** The two outcomes of interest in this article, death and CPEs, might occur due to screening, partly because of screening or not because of screening. Due to the scarce reporting of information that could be used to judge causality, we refrained from making any assumptions. In effect, we report occurrences of physical harm associated with CRCSPs.

## 2.9 Synthesis of results

**2.9.1. Combining data across studies.** We promoted homogenous studies in meta-analyses by categorising and subcategorising harms. We combined results from studies irrespective of their study design, i.e. combining randomised and non-randomised studies, because studies

were one-armed, i.e. the entire study population receives the intervention, there is no control group, and NRSs provide equally valid results concerning the harmful effects of interventions compared to RCTs [45]. As recommended in ROBINS-I's guidance, our primary outcomes are findings from studies that report the follow-up time without critical risk of bias [40]. We estimated a weighted and unweighted distribution of the risk of bias for each of the four screening interventions for each bias domain. We calculated the weight of each study as the size of its study population divided by the size of the total population across studies [46].

## 2.10 Statistical analysis and presentation of findings

Meta-analyses used Poisson regression models with a random effect of study and population size as an offset. Heterogeneity was quantified using $I^2$, $\tau^2$, and $\chi^2$ [47]. We calculated the study-specific confidence intervals using the Clopper-Pearson method [48]. We used Microsoft Excel [49] for descriptive statistics and R [50] for analyses. We investigated whether findings regarding harm from studies with a critical risk of bias differed from those without a critical risk of bias since such trends have been found in a related systematic review [51]. Polypectomy increases the risk of adverse events [4, 52, 53]. Therefore, we report the number of studies that provided polypectomies and the rate of polypectomies in these studies. Findings from meta-analyses are reported per 100,000 people screened with 95% confidence intervals. Findings are reported corresponding to the seven aims of the review with the aim in parentheses in the subtitle, e.g. aim 3 (A3).

## 3. Results

### 3.1 Study selection

The search strategy identified 23,281 publications after the removal of duplicates. Of these, 134 publications were included for review (Fig 2) (Appendix 8 in S1 File). We identified 12 ongoing studies (Appendix 9 in S1 File).

We analysed eight publications separate from other studies: three studies due to their design, precluding comparison to other studies [54–56], and five studies due to providing data about harms from an unscreened control group (Appendix 10 in S1 File). Findings from the screened group in the five studies with a control group were analysed with the other studies. The 131 publications reported results from 151 distinct subpopulations.

### 3.2 Characteristics of subpopulations with assessment of CPEs or deaths

The 151 subpopulations were from studies conducted in 23 different countries. Deaths or CPEs were assessed for 80 subpopulations (53%), 23 subpopulations were from RCTs (29%), and 57 subpopulations were from NRS (71%) (Appendix 11 in S1 File). Study populations in RCTs had an age span between 45 to 75 years of age, and the percentage of women ranged from 37% to 54%. Sociodemographic information was seldom reported (22%). Key characteristics of subpopulations with assessments of deaths and CPE are available in Appendices 12 and 13 in S1 File.

### 3.3 Types of physical harm associated with CRCSPs (A1)

Across the 134 publications included for review, 53 (40%) provided an explicit conceptualisation of what constituted physical harms associated with CRCSPs, with 44 different definitions across publications (Appendix 14 in S1 File). We categorised the evidence into 17 types of physical harm associated with CRCSPs:

# PRISMA FLOW CHART

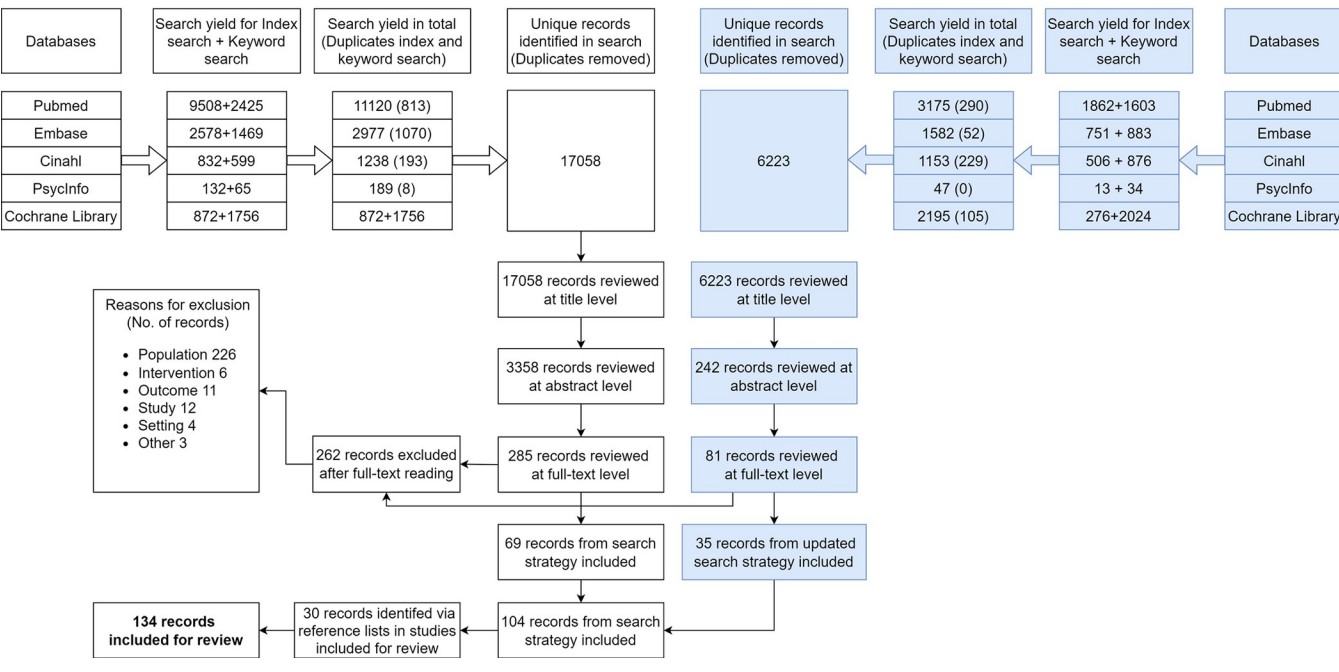

**Fig 2. Study selection process (PRISMA flow diagram).**

1. Death

2. Perforation

3. Cardiopulmonary events

4. Bleeding

5. Post-polypectomy syndrome

6. Infections

7. Inflammatory complications

8. Colorectal symptoms

9. Sedation-related complications

10. Complications associated with bowel preparation

11. Sleep disturbances

12. Nausea/vomiting

13. Dizziness

14. Pain

15. Discomfort

16. Other harms, e.g. hospitalisation, major morbidity without further specification

17. Other harms (Miscellaneous)

**Table 1. Adequacy of harm measurement in studies with assessment of deaths.**

| Procedure | All (N) | % | FS (N) | % | TCfobt (N) | % | TConly (N) | % | TCfollowup (N) | % |
|---|---|---|---|---|---|---|---|---|---|---|
| Subpopulations receiving procedure | 151 | 100 | 29 | 19 | 52 | 34 | 45 | 30 | 19 | 13 |
| Subpopulations with outcome assessed* | 39 | 26 | 3 | 10 | 16 | 31 | 14 | 31 | 6 | 32 |
| Follow-up time reported** | 39 | 100 | 3 | 100 | 16 | 100 | 14 | 100 | 6 | 100 |
| Outcome assessor reported** | 14 | 36 | 1 | 33 | 7 | 44 | 4 | 29 | 2 | 33 |
| Measurement tool reported** | 32 | 82 | 2 | 67 | 13 | 81 | 12 | 86 | 5 | 83 |

Abbreviations: Number of subpopulations (N), Percentage (%)

* Percentage calculated as the number of subpopulations with outcome assessment/number of subpopulations receiving the procedure, e.g. 39/151.

** Percentage calculated as the number of subpopulations with the given variable, e.g. follow-up time reported/the number of subpopulations with outcome assessment, e.g. 39/39 = 100%.

### 3.4 Adequacy of harm measurement and reporting across studies (A2)

**Death.** Across the four screening interventions, the outcome "death" was explicitly assessed for 57 of the 151 (38%) subpopulations. For 39 of these 57 subpopulations, the follow-up time was reported (68%). For these subpopulations, details about the outcome assessor were reported in 36% of studies, and the measurement method was reported in 82% of studies (Table 1).

**Cardiopulmonary events.** Across the four screening interventions, 113 CPEs of any type were assessed for 55 of the 151 subpopulations (36%). Of note, for 104 of the 113 subpopulations (92%) with one or more occurrences of CPEs of any type, only 27% of assessments included information about the consequences of the outcome (Table 2).

### 3.5 Modifiers on the risk of harm or the consequences thereof (A3)

Factors potentially modifying the risk of deaths or CPEs and the consequences of the latter were too few and too heterogeneous to make meaningful interpretations of the data.

### 3.6 Characteristics of the risk of bias of findings (A4)

**3.6.1. Risk of bias in assessments of death.** None of the 57 subpopulations with an assessment of deaths had a low risk of bias in any bias domain. Three (5%) subpopulations had a

**Table 2. Adequacy of harm measurement in studies with assessment of cardiopulmonary events.**

| Procedure | All | % | FS | % | TCfobt | % | TConly | % | TCfollowup | % |
|---|---|---|---|---|---|---|---|---|---|---|
| Subpopulations receiving procedure | 151 | 100 | 29 | 19 | 54 | 36 | 48 | 32 | 20 | 13 |
| Number of CPEs assessed | 113 | 100 | 13 | 12 | 27 | 24 | 67 | 59 | 6 | 5 |
| **Appraisal of harm reporting** | | | | | | | | | | |
| | N | % | N | % | N | % | N | % | N | % |
| Follow-up time* | 88 | 78 | 6 | 46 | 23 | 85 | 56 | 84 | 3 | 50 |
| Outcome assessor* | 45 | 40 | 3 | 23 | 18 | 67 | 23 | 34 | 1 | 17 |
| Measurement tool* | 79 | 70 | 4 | 31 | 26 | 96 | 44 | 66 | 5 | 83 |
| No. subpopulations with > 0 events* | 104 | 92 | 12 | 92 | 25 | 93 | 62 | 93 | 5 | 83 |
| Information about the consequence of harm** | 31 | 27 | 3 | 23 | 3 | 11 | 25 | 37 | 0 | 0 |

Abbreviations: Number of subpopulations (N), Percentage (%)

*Proportion calculated from the number of assessments of cardiopulmonary events in total, i.e., N = 113.

**Proportion calculated from the number of the number of subpopulations with > 0 events

moderate risk of bias as the worst score [57–59]. The worst bias score was either serious or critical for the remaining 54 subpopulations (95%). The most frequent cause for critical risk of bias was measurement bias (46% of subpopulations) (Fig 3 in S2 File). When bias assessments were weighed according to the size of study populations, 97% of the evidence was at serious or critical risk of bias (Fig 4 in S2 File).

**3.6.2. Risk of bias in studies that assessed CPEs.** None of the 113 assessments of CPEs had a low risk of bias in any bias domain. The worst bias score was moderate in six assessments (5%). The worst bias score was either serious or critical in 107 (95%) assessments. The most frequent cause for critical risk of bias was measurement bias, occurring in 22% of assessments (Fig 5 in S2 File). When bias assessments were weighed according to the size of study populations, 99% of the evidence was either at serious or critical risk of bias (Fig 6 in S2 File).

## 3.7 Characteristics of the certainty of the evidence (A5)

**3.7.1. GRADE assessments of findings on deaths.** None of the quantitative analyses had a GRADE rating above very low certainty, with further downgrading in all analyses. All analyses were downgraded -1 due to the likelihood of publication bias and -2 due to serious risk of bias in > 50% of studies (Appendix 17 in S1 File).

**3.7.2. GRADE assessments of findings on CPEs.** None of the quantitative analyses had a GRADE rating above very low certainty, with further downgrading for all subcategories of CPEs for all four screening interventions (Appendix 18 in S1 File). All analyses were downgraded -1 due to the likelihood of publication bias. We downgraded the evidence with -2 due to serious risk of bias in 25 analyses (89%) and with -1 in two analyses 7%). Only one analysis (4%) was not rated down due to the risk of bias (Vasovagal reaction short-term associated with TConly). Regarding inconsistency, 13 analyses (46%) were assessed for inconsistency, with six analyses downgraded -2 (46%), one analysis downgraded -1 (8%) and six analyses not downgraded (46%). The most frequent reasons for downgrading due to inconsistency were differences between studies regarding outcome measurement, outcome assessor, the expertise of endoscopists, whether sedation was provided and the rate of polypectomies (Appendix 18 in S1 File). We downgraded due to imprecision, with -1 in nine analyses (32%) and -2 in one analysis (4%). Due to the criterion "large effect size", we upgraded the evidence with +1 in four analyses (14%).

## 3.8 Occurrences of death (A6)

Death was assessed for 57 (38%) of 151 subpopulations. Of these, 26 (46%) had a critical risk of bias (Appendix 15 in S1 File). Deaths were analysed as the total number of deaths, i.e. deaths with any follow-up time (Death-AFU), which was further subcategorized into two subcategories: Deaths with follow-up time reported (Death FUR) and deaths without follow-up time reported (Death NRFU) (Table 3). For the 39 subpopulations with reporting of follow-up time (68% of studies), the follow-up time was 30 days in 32 studies (82%) and between seven days and three months in the seven other studies (18%). We did not find systematic differences between the risk of deaths and various lengths of follow-up time included in studies.

**3.8.1. Sigmoidoscopy and deaths.** Death was assessed for seven subpopulations (24%), with 202,933 people screened across the studies. Polypectomy was performed for all seven subpopulations (100%) with a weighted average of 24 polypectomies per 100 people screened. Across studies, 17 cases of death were reported, with the number of deaths per 100,000 people screened ranging from 0 to 187 (Figs 7–9 in S2 File)

**3.8.2. Colonoscopy following FOBT and deaths.** Death was assessed for 20 subpopulations (37%), with 830,233 people screened across the studies. The weighted average rate of

**Table 3. Meta-analyses of deaths associated with colorectal cancer screening programmes.**

|  | FS | TConly | TCfobt | TCfollowup |
|---|---|---|---|---|
| **Death-AFU, total** | 6, [1–45], (28%), 7S, N = 202,801, GS: -4 | 1, [0–10] (92%), 21S, N = 4,693,743, GS: -2 | 3, [1–9] (69%), 22S, N = 859,336, GS: -2 | 7, [0–95] (0%), 7S, N = 26,059, GS: -5 |
| • critical | 9, [5–16] (51%), 5S, N = 126,071, GS: NA | 0, [0–18] (92%), 11S, N = 3,145,336, GS: NA | 1, [0–7] (0%), 6S, N = 106,923, GS: NA | 3, [0–3000] (0%), 4S, N = 17,097, GS: NA |
| • non-critical | 3, [0–87] (0%), 2S, N = 76,739, GS: -4 | 5, [1–32] (91%), 10S, N = 1,548,407, GS: -2 | 5, [2–13] (75%), 16S, N = 752,413, GS: -2 | 11, [2–79] (0%), 3S, N = 8,962, GS: -5 |
| **Death-FUR, total** | 8, [3–20] (0%), 3S, N = 184,443, GS: -4 | 9, [3–31] (88%), 14S, N = 1,389,406, GS: -5 | 4, [1–11] (75%), 16S, N = 756,854, GS: -5 | 9, [1–93] (0%), 6S, N = 23,313, GS: -5 |
| • critical | 9, [4–17] (-%), 1S, N = 107,704. GS: NA | 0, [0–1486] (0%), 7S, N = 177,936, GS: NA | 0, [0–10] (0%), 2S, N = 37,646, GS: NA | 9, [0–421] (0%), 3S, N = 14,351, GS: NA |
| • non-critical | 3, [0–87] (0%), 2S, N = 76,739, GS: -4 | 23, [10–55] (94%), 7S, N = 1,211,470, GS: -5 | 5, [2–13] (78%), 14S, N = 719,208, GS: -5 | 11, [2–79] (0%), 3S, N = 8,962, GS: -5 |
| **Death-NRFU, total** | 8, [1–81] (0%), 4S, N = 18367, GS: NA | 0, [0–0] (0%), 7S, N = 3,304,337, GS: -3 | 2, [0–14] (38%), 6S, N = 102,482, GS: -4 | 0, [0–134] (98%), 1S, N = 2,746, GS: NA |
| • critical | 8, [1–81] (0%), 4S, N = 18367, GS: NA | 0, [0–0] (0%), 4S, N = 2,967,400, GS: NA | 1, [0–10] (0%), 4S, N = 69,277, GS: NA | 0, [0–134] (98%), 1S, N = 2,746, GS: NA |
| • non-critical | NA | 0, [0–1] (0%), 3S, N = 336,937, GS: -3 | 6, [0–79] (0%), 2S, N = 33,205, GS: -4 | NA |

Each cell should be read as follows: The number, e.g. 6, is the weighted average of deaths per 100,000 people screened, followed by a 95% confidence interval, e.g. [1–45], the $I^2$ measure of heterogeneity, e.g. (28%), the number of subpopulations in the analysis, e.g. 7S, and how many people that corresponds to, e.g. N = 202,810 people, and finally the GRADE Score for the pool of subpopulations that contribute with data for the outcome, e.g. GS: -4.

Critical: Subpopulations with a critical risk of bias for the outcome

Non-critical: Subpopulation with low to serious risk of bias for the outcome

Abbreviations: Subpopulations (S), Not Applicable (NA), GRADE score (GS), summed estimates of deaths with any follow-up time in studies (Death-AFU), summed estimates of deaths with follow-up time reported in studies (Death-FUR), and summed estimates of deaths without follow-up time reported in studies (Death-NRFU).

polypectomy was 57%. Across studies, 83 cases of death were reported. The number of deaths ranged from 0 to 95 per 100,000 people screened across studies (Figs 10–12 in S2 File)

**3.8.3. Once-only colonoscopy and deaths.** Death was assessed for 21 subpopulations (44%), with 4,658,351 people screened across the studies. The weighted average rate of polypectomy was 36%. Across studies, 220 cases of death were reported, ranging from 0 to 94 per 100,000 people screened (Figs 13–15 in S2 File).

**3.8.4. Colonoscopy following various screening tests and deaths.** Death was assessed for seven subpopulations (35%), with 26,139 people screened across the studies. The weighted average rate of polypectomy was 44%. Across studies, 4 cases of death were reported, ranging from 0 to 60 per 100,000 people screened (Figs 16, 17 in S2 File).

## 3.9 Risk of Cardiopulmonary events (A6)

There were 113 assessments of CPEs of any type across the 55 subpopulations with an assessment of CPEs (36%), which we categorised into seven types of CPE with either long or short-term follow-up, amounting to 17 distinct subcategories (Appendix 6 in S1 File). We subcategorised according to the type of CPE, e.g. arrhythmia or heart failure, and according to the included follow-up time, dichotomised into a) short-term (< 14 days) and b) long-term (30 days). Outcomes in the categories "Follow-up time not reported", "other", and "Non-defined cardiopulmonary events (NDCPE)" accounted for 41% of the evidence about CPEs and were excluded from further analysis due to unclear reporting and undefined events). We did meta-analyses of the remaining 14 subcategories of CPE (Table 4). Bias assessments for the 113 assessments of CPEs are available in (Appendix 16 in S1 File).

**Table 4. Meta-analyses of cardiopulmonary events associated with colorectal cancer screening programmes.**

| | FS | TConly | TCfobt | TCfollowup |
|---|---|---|---|---|
| **ACS short-term** | - | 21, [1–514] (66%), 3S, N = 14,383, GS: -5 | 9, [4–18] (0%), 1S, N = 78,831, GS: -3 | - |
| • critical | - | NA | NA | - |
| • non-critical | - | 21, [1–514] (66%), 3S, N = 14,383, GS: -5 | 9, [4–18] (0%), 1S, N = 78,831, GS: -3 | - |
| **Arrhythmia short-term** | - | 188 [116–287] (0%), 1S, N = 11,163, GS: -3 | 5, [1–13] (0%), 1S, N = 78,831, GS: -3 | - |
| • critical | - | NA | NA | - |
| • non-critical | - | 188 [116–287] (0%), 1S, N = 11,163, GS: -3 | 5, [1–13] (0%), 1S, N = 78,831, GS: -3 | - |
| **Heart failure short-term** | - | - | 1087, [28–5908] (0%), 1S, N = 92, GS: -2 | - |
| • critical | - | - | NA | - |
| • non-critical | - | - | 1087, [28–5908] (0%), 1S, N = 92, GS: -2 | - |
| **Pulmonary event short-term** | - | 65, [31–137] (99%), 4S, N = 1,373,976, GS: -5 | 1, [0–7] (0%), 1S, N = 78,831, GS: -3 | - |
| • critical | - | NA | NA | - |
| • non-critical | - | 65, [31–137] (99%), 4S, N = 1,373,976, GS: -5 | 1, [0–7] (0%), 1S, N = 78,831, GS: -3 | - |
| **Stroke short-term** | - | 42, [19–93] (0%), 2S, N = 14,284, GS: -4 | 3, [0–9] (0%), 1S, N = 78,831, GS: -3 | - |
| • critical | - | NA | NA | - |
| • non-critical | - | 42, [19–93] (0%), 2S, N = 14,284, GS: -4 | 3, [0–9] (0%), 1S, N = 78,831, GS: -3 | - |
| **TE short-term** | - | 45 [15–104] (0%), 1S, N = 11,163, GS: -3 | 3, [0–9] (0%), 1S, N = 78,831, GS: -3 | - |
| • critical | - | NA | NA | - |
| • non-critical | - | 45 [15–104] (0%), 1S, N = 11,163, GS: -3 | 3, [0–9] (0%), 1S, N = 78,831, GS: -3 | - |
| **Vasovagal reaction short-term** | 774, [336–1784] (98%), 2S, N = 16,879, GS: -5 | 180, [51–633] (95%), 2S, N = 23,075, GS: -1 | - | 900, [240–1570], 1S, N = 775, GS: -2 |
| • critical | NA | 72, [31–141] (0%), 1S, N = 11,163, GS: NA | - | NA |
| •- non-critical | 774, [336–1784] (98%), 2S, N = 16,879, GS: -5 | 428, [319–563] (0%), 1S, N = 11,912, GS: -1 | - | 900, [240–1570], 1S, N = 775, GS: -2 |
| **ACS long-term** | 14, [4–52] (84%), 2S, N = 148,378, GS: -3 | 49, [20–118] (98%), 8S, N = 778,541, GS: -6 | 22, [9–54] (0%), 3S, N = 22,322, GS: -4 | - |
| • critical | 31, [21–43] (-%), 1S, N = 107,704, GS: NA | 8, [1–54] (0%), 2S, N = 13,084, GS: NA | NA | - |
| • non-critical | 5, [1–18] (-%), 1S, N = 40,674, GS: -3 | 71, [32–157] (98%), 6S, N = 765,457, GS: -6 | 22, [9–54] (0%), 3S, N = 22,322, GS: -4 | - |
| **Arrhythmia long-term** | - | 176, [58–533] (99%), 6S, N = 758,075, GS: -6 | 21, [4–113] (0%), 2S, N = 19,338, GS: -3 | 30, [0–90], 1S, N = 3215, GS: NA |
| • critical | - | 78, [2–436] (-%), 1S, N = 1,276, GS: NA | NA | 30, [0–90], 1S, N = 3215, GS: NA |
| • non-critical | - | 204, [60–689] (99%), 5S, N = 756,799, GS: -6 | 21, [4–113] (0%), 2S, N = 19,338, GS: -3 | NA |
| **Heart failure long-term** | - | 201, [105–383] (99%), 4S, N = 753,603, GS: -6 | - | - |
| • critical | - | NA | - | - |
| • non-critical | - | 201, [105–383] (99%), 4S, N = 753,603, GS: -6 | - | - |

*(Continued)*

**Table 4.** (Continued)

| | FS | TConly | TCfobt | TCfollowup |
|---|---|---|---|---|
| **Pulmonary event long-term** | - | 204, [193–215] (0%), 2S, N = 646,371, GS: -3 | 11, [0–61] (0%), 1S, N = 9,061, GS: -3 | - |
| • critical | - | 78, [2–436] (-%), 1S, N = 1,276, GS: NA | NA | - |
| • non-critical | - | 204, [193–215] (-%), 1S, N = 645,095, GS: -3 | 11, [0–61] (0%), 1S, N = 9,061, GS: -3 | - |
| **Stroke long-term** | - | 58, [48–69] (76%), 5S, N = 1,303,320, GS: -4 | 134, [37–343] (0%), 1S, N = 2,984, GS: -3 | - |
| • critical | - | 0, [0–289] (-%), 1S, N = 1,276, GS: NA | NA | - |
| • non-critical | - | 58, [49–70] (82%), 4S, N = 1,302,044, GS: -4 | 134, [37–343] (0%), 1S, N = 2,984, GS: -3 | - |
| **TE long-term** | - | 77, [53–114] (0%), 2S, N = 33,639, GS: -3 | 18, [8–41] (0%), 4S, N = 32,599, GS: -3 | - |
| • critical | - | NA | NA | - |
| • non-critical | - | 77, [53–114] (0%), 2S, N = 33,639, GS: -3 | 18, [8–41] (0%), 4S, N = 32,599, GS: -3 | - |
| **Vasovagal reaction long-term** | - | 820, [168–4012] (99%), 3S, N = 81,261, GS: -5 | 0, [0–49] (0%), 1S, N = 7,467, GS: -3 | - |
| • critical | - | NA | NA | - |
| • non-critical | - | 820, [168–4012] (99%), 3S, N = 81,261, GS: -5 | 0, [0–49] (0%), 1S, N = 7,467, GS: -3 | - |

Each cell should be read as follows: The number, e.g. 820, is the weighted average of vasovagal events with long-term follow-up per 100,000 people screened, followed by a 95% confidence interval, e.g. [168–4012], the $I^2$ measure of heterogeneity, e.g. (99%), the number of subpopulations in the analysis, e.g. 3S, and how many people that corresponds to, e.g. N = 81,261 people, and finally the GRADE Score for the pool of subpopulations that contribute with data for the outcome, e.g. GS: -5.

Critical: Subpopulations with a critical risk of bias for the outcome

Non-critical: Subpopulation with low to serious risk of bias for the outcome

Abbreviations: No studies for analysis (-), Subpopulations (S), Not Applicable (NA), GRADE score (GS), summed estimates of deaths with any follow-up time in studies (Death-AFU), summed estimates of deaths with follow-up time reported in studies (Death-FUR), and summed estimates of deaths without follow-up time reported in studies (Death-NRFU), Acute Coronary Syndrome (ACS), Thromboembolic event (TE).

**Consequences of CPEs.** One or more CPE events occurred in 104 subpopulations (92%). Here, consequences of CPE were reported for 31 subpopulations (27%). For 19 subpopulations (17%), it was reported whether people experiencing a CPE were hospitalised without any details about the duration of hospitalisation, complicating issues or treatments given. The severity of the CPE was reported for ten subpopulations (9%). One study accounted for six of these assessments, reporting whether CPEs were mild, moderate, severe or fatal, using the ASGE lexicon terminology [60].

**3.9.1. Sigmoidoscopy and cardiopulmonary events.** Cardiopulmonary events were assessed for ten of the 29 subpopulations (34%), with 13 assessments of CPEs. Of these, nine assessments could not be used for analyses (69%) due to being categorised in the three categories mentioned above. The remaining assessments ranged from vasovagal reaction with short-term follow-up to acute coronary syndrome with long-term follow-up. Vasovagal reaction short-term was assessed for two subpopulations with 16,879 people screened: 140 events occurred, ranging from 424 to 1406 events per 100,000 people screened with a weighted average of 774 events per 100,000 people screened [336–1784] (Fig 18 in S2 File). ACS long-term was assessed for two subpopulations with 148378 people screened, 35 events occurred, ranging from 5 to 31 events per 100,000 people screened with a weighted average of 14 events per 100,000 people screened [4–52] (Fig 19 in S2 File).

**3.9.2. Colonoscopy following FOBT and cardiopulmonary events.** Cardiopulmonary events were assessed for 14 of the 54 subpopulations (26%) receiving colonoscopy after FOBT, with 27 CPEs reported. Of these, nine assessments could not be used for analyses (33%) due to being categorised in the three categories mentioned above. For the remaining assessments, three subcategories of CPE (18%) were assessed for more than two subpopulations: ACS long-term, arrhythmia long-term and TE long-term (sTable 4 in S1 File). Meta-analyses illustrate the weighted average rate of each subcategory of CPE (Figs 20–22 in S2 File).

**3.9.3. Once-only colonoscopy and cardiopulmonary events.** Cardiopulmonary events were assessed for 26 of the 48 (54%) subpopulations receiving a once-only colonoscopy, with 67 CPEs reported. Of these, 24 assessments could not be used for analyses (36%) due to being categorised in the three above mentioned categories. Of the remaining assessments, 11 subcategories of CPE (65%) were assessed for more than two subpopulations (sTable 4 in S1 File). Meta-analyses illustrate the weighted average rate of each subcategory of CPE (Figs 23–33 in S2 File).

**3.9.4. Colonoscopy following any type of screening test and cardiopulmonary events.** Cardiopulmonary events were assessed for five of the 20 subpopulations (25%) receiving colonoscopy following any screening test other than FOBT, with 6 CPEs reported. Four assessments could not be used for analyses (67%) due to being categorised in the three categories mentioned above, i.e. "Follow-up time NR", "other", and "NDCPE". None of the other subcategories of CPE was assessed for two or more subpopulations.

## 4. Discussion

### 4.1 Summary of main findings

We included 134 publications for review, of these 8 (6%) were only available in abstract form. Our findings suggest that physical harms associated with CRCSPs can be categorised into 17 types of physical harm. We found that the definition of physical harm as an overall concept varied substantially, with 44 different definitions across the studies. Most studies did not define outcomes clearly and lacked information about follow-up time, outcome assessment methods, and the consequences, i.e. severity of harm. None of the 18 RCTs (64%) published after the CONSORT-harms extension referred to this guideline [61, 62].

Only one-third of studies included for review assessed the two most serious types of physical harm associated with CRCSPs, i.e. deaths and CPEs. Meta-analyses indicate that across the four screening procedure groups, 3 to 23 deaths occur during CRCSPs per 100,000 people screened. Yet, confidence intervals span from 0 to 87 deaths per 100,000 people screened in analyses of non-critical studies with follow-up time (primary outcome), illustrating uncertainty about the true effect. The evidence on CPEs was very heterogeneous, necessitating the subcategorization of CPEs into 17 subcategories. Here, 41% of the evidence was too poorly measured or reported to allow quantification. Most subcategories of CPEs were assessed for once-only colonoscopy, ranging from 0 to 4012 events per 100,000 people screened, depending on the type of CPE. Adding to the heterogeneity of the evidence, there was a very high risk of bias in findings both from RCTs and NRS, with a serious or critical risk of bias of findings about deaths and CPEs in 95% of assessments. Of note, we found a trend towards low-quality studies underestimating the risk of both deaths and CPEs. This is problematic since other reviews have found that low-quality studies also overestimate screening benefits [51]. As for overall trustworthiness, the GRADE rating was very low in all analyses, with further downgrading between -2 to -7. Inadequate reporting of harms prevented meaningful analyses of the severity and consequences of CPEs. Poor reporting also impeded the assessment of whether characteristics of screening intervention delivery or characteristics of study populations, e.g.

age or gender, potentially modified the risk or the consequences of physical harm. However, we found that arrhythmia, vasovagal events, heart failure, and acute coronary syndrome, occurred at higher rates among older people and if polypectomy is performed, which is consistent with findings in other studies [63] and in other reviews [64], potentially caused by the greater comorbidity in this age group, which have led to debate about the appropriate cut-off age for when to stop screening [65, 66].

## 4.2 Strengths and limitations

The review adhered to the best available guidance for systematic reviews of adverse events of medical interventions [29, 33, 34, 67]. The search strategy identified 77 publications (57%) not formerly included in other reviews, which makes this review the most comprehensive account of the potential physical harms of CRCSPs to date. One of the review's main limitations is that it exclusively accounts for physical harms resulting from invasive diagnostic procedures, e.g. sigmoidoscopy and colonoscopy, and not physical harms associated with procedures or treatment before or after these procedures. In effect, the true extent of physical harm associated with CRCSPs requires analyses of the other steps of the screening cascade, e.g. surgical treatment following diagnostic workup. In addition, we must acknowledge that this publication only provides an account of the two major types of physical harm, namely deaths and CPEs, related to the diagnostic procedures during screening. Analyses of other types of physical harm associated with CRCSPs are planned for separate publications. Based on the literature in the area, we judged that the benefits of separating reporting of the review's findings into more than one publication outweigh the caveats and ethical concerns of this practice [68, 69]. Another limitation within the best-available evidence and hence our findings, is that only 5 studies (4%) had an unscreened control group with measurement of the outcomes of interest, physical harm. In effect, findings about physical harms, e.g. deaths and cardiopulmonary events, are susceptible to bias from confounding, i.e. events happening during or in the 30 days following diagnostic procedures but not because of these procedures. This acts to bias our findings towards the overestimation of harms related to diagnostic procedures. However, more mechanisms likely act to cause underestimation of harms related to screening (section 5.4 Implications of findings). Also, we judged that an outcome was not assessed when it was not reported. It might be that physical harm was assessed but not reported due to zero findings. However, we judged that the opposite was likely more often the case: zero events occurred due to inadequate assessment.

 A major limitation concerning the evidence at hand is the heterogeneity of studies in the area: we observed large variance in how harms were defined, which harms were measured, how these were measured and finally how harm assessments were reported. The heterogeneity of the evidence made the task of dividing outcomes into outcome types, and further to subcategorize each type of harm according to severity and follow-up time to make quantifiable groups of outcomes, a difficult task. It might be argued that our 17 types of harm, and how these were subcategorized, are to some extent arbitrary. However, we used the best available guidance in the area to inform our categorisations, which were also thoroughly discussed in the author group. Still, our list of outcomes, and methods for subcategorization, should be seen as a work in progress that future studies might improve upon. Another implication of the large heterogeneity of the evidence included for review concerns the legitimacy of compiling evidence in meta-analyses. Although studies were heterogeneous, we believe that our meta-analyses were justified due to a) stringent eligibility criteria of the review, b) dividing analyses according to the four types of screening interventions, c) systematic categorisation of outcomes, d) stratification of outcomes into three causality categories, e) exclusion of studies with critical risk of

bias from analyses, and f) use of mixed effects meta-analytic models. Of note, to allow compilation of the evidence we had to accept some heterogeneity between study characteristics, e.g. the age span of study populations, and within the categories of harm, to avoid ending up with as many categories as there were studies included for review. For example, in the outcome category deaths with follow-up time (Death-FUR) we accepted that the follow-up time varied from during the procedures to 3 months after the procedure also because the large majority of assessments (82%) had a follow-up time of 30 days. These spans of heterogeneity for a certain variable, here follow-up time, was allowed when we did not find significant variation in the occurrence of the outcome, i.e. despite different follow-up times between assessments we did not observe any systematic differences. Finally, we found that even with substantial amendments to the GRADE approach (Appendix 5 in S1 File), it is not sufficiently tailored to provide useful assessments of the certainty of the often poor quality and heterogeneous evidence on adverse events. We found that the tool reached a "floor effect", i.e. all studies had low certainty, which hindered the tool's usability from differentiating questionable, yet useful, evidence from useless or even misleading evidence. Also, despite having made topic-specific guidance to use the GRADE approach (Appendix 5 in S1 File), we experienced ambiguity in decisions about up- and downgrading the evidence, highlighting the need for clearer criteria in the context of adverse events, heterogeneous evidence and one-armed trials. Of note, these concerns are not evident from the current guidance in the GRADE handbook [41].

## 4.3 Comparison to other systematic reviews in the area

We found 10 systematic reviews assessing physical harm associated with CRCSPs [4, 11–19] (Appendix 19 in S1 File). Estimates of death associated with CRCSPs varied significantly across reviews, from less than one event to 25 events per 100,000 people screened, which was a smaller interval than our findings with 0 to 87 deaths per 100,000 people screened across the four screening interventions (Appendix 20 in S1 File). None of the other ten reviews quantified any of the 14 types of CPEs that we deemed were quantifiable from the 17 categories made in our review. Most former systematic reviews highlighted issues with evidence similar to those found in this review, including heterogeneous and inadequate harm assessment methods and poor reporting of harm in clinical studies. However, former systematic reviews differed substantially concerning how they conceptualised what constitutes physical harm associated with CRCSPs, e.g. none of the ten reviews had the same definition of physical harm.

Further, many existing reviews restricted the definition of harm to serious adverse events. Applying this restriction to our findings would lead to 12 (71%) of the 17 types of physical harm associated with CRCSPs identified in this review not counting as harm. All reviews claimed to assess the harms of screening, i.e. implicitly, the entire screening cascade. Yet, none of the existing reviews in the area, including ours, have included a harm assessment of all steps of the screening cascade, e.g. including harms of bowel preparation and surgery following the detection of cancer (Appendix 19 in S1 File). Of note, a recent review of the harms of CRCSPs [4] found that the certainty of the evidence was moderate, which is in stark contrast to our findings (Appendices 17 and 18 in S1 File).

## 4.4 Implications of findings

Due to the abovementioned issues with the evidence on CRCSPs and screening in general, we believe that our findings are conservative for the following reasons: 1) the harms of screening tend to be underreported in general [23, 45], 2) most studies had either serious or critical risk of bias in one or more of the six bias domains assessed, and all six types of bias most likely draw the reported rate of harm towards the null, 3)when authors stated that a harmful event

was not considered related to the screening procedure we did not include this harm, making it likely that we miss events that actually were related to screening,, 4) the minority of included studies that assessed ongoing screening programmes consistently found more types and greater severity of harms of CRCSPs compared to reported rates of physical harm in clinical studies [58, 60, 70], which due to their weight in meta-analyses draw the collective estimate towards the null, 5) publication bias is more frequent in studies that report adverse events because they are more difficult to publish [45, 67] and many aspects of the evidence included for review points to probable publication bias, e.g. 8 publications (6%) included for review were only available in abstract form and were likely never published as full articles, many small-scale studies were included for review, and the majority of included studies were observational, known to be more prone to publications bias. We find it likely that publication bias would act to bias results towards the null, causing underestimation of the true extent of harm (Appendix 5 in S1 File). Our findings highlight the need for consensus on how harms should be measured and reported in clinical studies, e.g. a core outcome set [71]. Currently, a taxonomy is lacking that clearly defines and conceptualises all effects of screening, e.g. benefits, harm, costs, and legal and ethical implications for people and society. Our taxonomy distinguishes between 17 categories of physical harm, which are further subcategorised, resulting from screening. Here, we define harm as any negative effect from screening participation perceived by the screening participant or their significant others [72, 73].

Dissemination of our findings to clinicians and laypeople, ideally via incorporating them into screening information materials, could help create an informed dialogue about the balance between the benefits and harms of screening. More emphasis on the potential harms of screening in information materials is warranted, especially in the light of evidence showing that laypeople and health professionals tend to overestimate the benefits and underestimate the harms of medical prevention [74, 75]. Current information materials potentially compound people's tendency to underestimate harms compared to the benefits of screening about screening, where harms are underreported or communicated unbalanced in comparison to benefits [26, 76–79]. However, a comprehensive, evidence-based information leaflet for citizens invited to screening will not solve all problems of informed choice. Due to the overestimation of benefits relative to the underestimation of harm, a perception gap can appear where laypeople are unable to comprehend the evidence due to strong pre-assumptions about screening [80].

### 4.5 Future research

Screening is a complex cascade of interlinked events with potential harm at each cascade step. For example, bowel preparation is part of the diagnostic work-up in CRCSPs. Yet, only one study (1%) included for review systematically assessed potential adverse events of bowel preparation [81], even though screening participants often rate bowel preparation as the most unpleasant part of the screening cascade [82–85]. Future reviews could assess harms resulting from other phases of CRCSPs, e.g. bowel preparation, surveillance programmes and preventive surgery. Such reviews could then be compiled in a meta-review to allow a fair comparison of the potential benefits of CRCSPs. Future clinical studies should focus on more rigorous measurement and more detailed reporting of the harms of screening. In addition, future studies could in general provide better reporting of study-related aspects, e.g. we noted inadequate reporting of funding sources, the number of screening rounds included in studies and more in most publications included for review. Recent studies have called for a standardized system for reporting harm, both in clinical studies [60] and in systematic reviews about screening programmes in general [86]. Our list of the 17 types of physical harm associated with CRCSPs

could provide a starting point for developing a Core Outcome Set to guide a more comparable and comprehensive assessment of potential physical harms associated with CRCSPs [71]. To further improve reporting of adverse events in general, publishers could demand that publications from RCTs comply with the CONSORT-harms statement. Further, they could support the development of an extension to the STROBE guideline to improve reporting of adverse effects in NRSs.

## 5. Conclusion

We found that the evidence on physical harms associated with CRCSPs is heterogeneous, has a high risk of bias, and that the certainty of the evidence is very low. We found that studies with a critical risk of bias tended to report lower estimates of harm, i.e., deaths and CPEs, than studies without a critical risk of bias. We found that most studies in the area had very limited scope for measuring and reporting harms, e.g. restricting definition to serious adverse events during the procedure and thus omitting many types of harm. In addition, most studies failed to provide a clear concept of what constituted adverse events, and harms were heterogeneously defined, inadequately measured and poorly reported.

We found that none of the 18 RCTs published after the publication of the CONSORT-harms extension referred to this guideline. Based on our findings, we can conclude that deaths during CRCSPs are rare, yet they do occur, but with very low certainty of the evidence. Also, many types of CPEs occur during CRCSPs, including arrhythmia, vasovagal events, heart failure, and acute coronary syndrome, especially for older people and if polypectomy is performed. Our findings are likely to be conservative; therefore, the physical harms of ongoing real-world CRCSPs are likely more frequent and more severe than reflected by the evidence we have compiled here in this review. Concerning future research, better evidence is needed, with adequate measurement and reporting of the potential physical harms of screening in studies, systematic reviews, and ongoing screening programmes.

## Supporting information

**S1 File. Appendices.**
(PDF)

**S2 File. Figures.**
(PDF)

**S3 File. PROSPERO protocol.**
(PDF)

**S4 File. PRISMA-HARMS checklist (completed).**
(PDF)

**S5 File. AMSTAR checklist (completed).**
(PDF)

## Acknowledgments

Thanks to the people listed below for their help in assessing the eligibility of studies published in languages other than Scandinavian and English.

We thank our colleagues listed below for their support in assessing studies' eligibility and translating relevant publications.

Prof. Dr. David Klemperer from Germany, affiliated with Ostbayerische Technische Hochschule Regensburg Faculty of Social and Health Care Sciences. Prof. MD. PhD Maciek

Godycki-Cwirko from Poland, affiliated with the Centre for Family and Community Medicine at the Medical University of Lpdz. MD, PhD, Janos Valery, from Brazil, affiliated with the Department of Preventive Medicine FMUSP. Dr. Giuseppe Febbo, from Italy, affiliated with Movimento Giotto. Pr. Dr. Jean Yes Le Reste from France, affiliated with EA 7479 SPURBO at université de bretagne occidentale in Brest. Mateja Bulc, MD, PhD, GP, is an Assoc. Prof. at the Department of Family Medicine at the Medical Faculty of Ljubljana University, Slovenia. Assoc. prof. Dr. Ph.D. Bohumil Seifert from the Czech Republic, affiliated with Charles University, First Faculty of Medicine, Department of General Practice in Prague. Prof. Ryuki Kassai from Japan, affiliated with the Department of Community and Family Medicine at Fukushima Medical University in Fukushima.

## Author Contributions

**Conceptualization:** Frederik Handberg Juul Martiny, John Brandt Brodersen.

**Data curation:** Frederik Handberg Juul Martiny, Anne Katrine Lykke Bie, Christian Patrick Jauernik, Or Joseph Rahbek, Sigrid Brisson Nielsen, Emma Grundtvig Gram, Isabella Kindt.

**Formal analysis:** Frederik Handberg Juul Martiny, Anne Katrine Lykke Bie, Christian Patrick Jauernik, Or Joseph Rahbek, Sigrid Brisson Nielsen, Emma Grundtvig Gram, Isabella Kindt, Volkert Siersma, Christine Winther Bang, John Brandt Brodersen.

**Funding acquisition:** Frederik Handberg Juul Martiny, John Brandt Brodersen.

**Investigation:** Frederik Handberg Juul Martiny, Anne Katrine Lykke Bie, Christian Patrick Jauernik, Or Joseph Rahbek, Sigrid Brisson Nielsen, Emma Grundtvig Gram, Isabella Kindt, Volkert Siersma, Christine Winther Bang, John Brandt Brodersen.

**Methodology:** Frederik Handberg Juul Martiny, Anne Katrine Lykke Bie, Christian Patrick Jauernik, Or Joseph Rahbek, Sigrid Brisson Nielsen, Emma Grundtvig Gram, Isabella Kindt, Volkert Siersma, Christine Winther Bang, John Brandt Brodersen.

**Project administration:** Frederik Handberg Juul Martiny, John Brandt Brodersen.

**Resources:** Frederik Handberg Juul Martiny, John Brandt Brodersen.

**Software:** Frederik Handberg Juul Martiny, Volkert Siersma, Christine Winther Bang.

**Supervision:** Volkert Siersma, John Brandt Brodersen.

**Validation:** Frederik Handberg Juul Martiny, Anne Katrine Lykke Bie, Christian Patrick Jauernik, Or Joseph Rahbek, Sigrid Brisson Nielsen, Emma Grundtvig Gram, Isabella Kindt, Volkert Siersma, Christine Winther Bang, John Brandt Brodersen.

**Visualization:** Frederik Handberg Juul Martiny, Volkert Siersma, Christine Winther Bang.

**Writing – original draft:** Frederik Handberg Juul Martiny.

**Writing – review & editing:** Frederik Handberg Juul Martiny, Anne Katrine Lykke Bie, Christian Patrick Jauernik, Or Joseph Rahbek, Sigrid Brisson Nielsen, Emma Grundtvig Gram, Isabella Kindt, Volkert Siersma, Christine Winther Bang, John Brandt Brodersen.

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
