## [Decision Letter · Decision Letter 0]

8 Mar 2023

PONE-D-22-34235Physical harms associated with sigmoidoscopy and colonoscopy during colorectal cancer screening - a systematic review with meta-analyses of deaths and cardiopulmonary eventsPLOS ONE

Dear Dr. Martiny,

Thank you for submitting your manuscript to PLOS ONE. After careful consideration, we feel that it has merit but does not fully meet PLOS ONE’s publication criteria as it currently stands. Therefore, we invite you to submit a revised version of the manuscript that addresses the points raised during the review process.

We look forward to receiving your revised manuscript.

Kind regards,

Divya Bhandari

Academic Editor

PLOS ONE

Journal Requirements: 

2. We note that you have referenced (ie. Bewick et al. [5]) which has currently not yet been accepted for publication. Please remove this from your References and amend this to state in the body of your manuscript: (ie “Bewick et al. [Unpublished]”) as detailed online in our guide for authors

Reviewers' comments:

Reviewer's Responses to Questions

**Comments to the Author**

1. Is the manuscript technically sound, and do the data support the conclusions?

Reviewer #1: Yes

Reviewer #2: Partly

Reviewer #3: Yes

Reviewer #4: Yes

Reviewer #5: Yes

Reviewer #6: Yes

Reviewer #7: Yes

2. Has the statistical analysis been performed appropriately and rigorously? 

Reviewer #1: Yes

Reviewer #2: I Don't Know

Reviewer #3: Yes

Reviewer #4: Yes

Reviewer #5: Yes

Reviewer #6: I Don't Know

Reviewer #7: Yes

3. Have the authors made all data underlying the findings in their manuscript fully available?

Reviewer #1: Yes

Reviewer #2: Yes

Reviewer #3: Yes

Reviewer #4: Yes

Reviewer #5: Yes

Reviewer #6: No

Reviewer #7: Yes

4. Is the manuscript presented in an intelligible fashion and written in standard English?

Reviewer #1: Yes

Reviewer #2: Yes

Reviewer #3: No

Reviewer #4: Yes

Reviewer #5: Yes

Reviewer #6: Yes

Reviewer #7: No

5. Review Comments to the Author

Reviewer #1: The authors presented a methodologically well-executed meta-analysis to quantify the risk of the most serious types of physical harm during CRCSPs, i.e., deaths and cardiopulmonary events (CPEs).

However, I believe that the manuscript can be improved and I present my suggestions for each section:

Abstract:

I suggest removing in line 30 "Design Systematic review with descriptive statistics and random-effects meta-analyses" putting it in Methods: Systematic review with descriptive statistics and random-effects meta-analyses carried out according to PRISMA. Systematic search in the literature for studies investigating physical harms associated...

On lines 35 and 36 I suggest that you exchange "We used the ROBINS-I tool and the GRADE approach to assess the internal and external validity of findings, respectively." para: We used the ROBINS-I tool and the GRADE approach to assess the risk of bias and certainty of evidence, respectively.

In lines 46 and 47 I suggest the change to "We found a tendency towards lower estimates of deaths and CPEs in studies with very low certainty of evidence and a high risk of bias compared to non-critical studies.

Methods

3.1 Study eligibility

The main inclusion and exclusion criteria defined by the authors for the selection of studies must be clearly described here as text and not indicating that they are in Appendix 1.

3.6 The external validity of findings across studies included for review

I suggest switching to:

3.6 The certainty of evidence of findings across studies included for review

Results

PRISMA 2020 recommends that in systematic reviews, authors present a main table, usually table 1, that shows the characteristics of the studies included in the review, containing authors, year, population, outcomes, main results and columns of risk of bias and GRADE certainty of evidence. Due to the large number of studies included, I believe that the authors can replace the confusing tables in the submitted manuscript with two tables, one with the characteristics of the studies in which the death outcome was evaluated and the other in which the CPEs outcome was verified.

On line 310 I suggest changing:

4.7 Characteristics of the external validity of findings (A5)

for:

4.7 GRADE certainty of evidence to outcomes (A5)

Table 3 in line 314 and table 4 in line 321 can be excluded and in the tables previously suggested by me for each study the authors present in the column GRADE certainty of evidence if the study was evaluated as very low, low, moderate and high, being able to represent by the circle with a positive sign according to GRADE handbook.

Discussion

In lines 581 to 583 of the conclusion the authors state "Also, many types of CPEs occur during CRCSPs, including arrhythmia, vasovagal events, heart failure, and acute coronary syndrome, especially for older people and if polypectomy is performed."

However, there is not at least one paragraph in the discussion that discusses hypotheses that explain why CPEs are more frequent in the elderly and in patients undergoing polypectomy, or whether these results are similar to those reported in other systematic reviews. I suggest you add a paragraph or two about this, as these are important results for clinical decision-making in screening the elderly population.

Conclusion

On lines 580 and 581 I suggest changing to:

Based on our findings, we can conclude that deaths during CRCSPs are rare, yet they do occur, but with very low certainty of evidence.

Reviewer #2: This manuscript is a very meaningful systematic review research about colorectal cancer screening and physical harms such as deaths and cardiopulmonary events. I agree with their conclusion that deaths during CRCSPs are rare, yet they do occur.

But I think that such serious harms were complications of endoscopic interventions like polypectomy and that endoscopic intervention should be excluded like surgical intervention.

Reviewer #3: Review Report

Title: Physical harms associated with sigmoidoscopy and colonoscopy during colorectal cancer screening - a systematic review with meta-analyses of deaths and cardiopulmonary events

Comments

1. The title should be very refined and shortened. E.g., time and design.

2. The type of the systematic review and meta-analysis should be specified.

3. Still the knowledge gap in this section is not too detail and the reasonings are weak,

4. Still the methods need some framing like;

The particular timing of the harm.

The setting, types of specific diagnosis, who diagnosed and how, the sources of data and its similarity and difference, the test of heterogeneity, the study period and the criteria of deciding that specific time, is that only general hospital or referral and specialized hospital, who verified and how? Needs to be assessed and presented well. in addition, the strength and limitation of the included studies should be also well assessed.

The philosophical stance of the researchers.

Results should be still logical, concise and short.

The discussion, conclusion and recommendation should be entirely drawn from the study objectives.

Regards,

Reviewer #4: Thanks for this opportunity to review this submission. The author of this study tried to systematically review and conduct a meta-analysis on the physical harms associated with colorectal cancer (CRC) screening interventions. The study is very well designed and conducted to investigate the interesting notion which appears also to be a crucial complication of CRC screenings. The manuscript is also well drafted. Just some revisions might help to enhance the draft. My comments and suggestions are provided below.

1. Lines 22-51: the abstract needs to be formatted according to the PLOS One guidelines for submission.

2. Lines 50-51: the final conclusion is somehow vague indicating “Physical harm during CRCSPs likely occurs more often than previously recognized”. A revision on the final conclusion might be better for abstract.

3. Lines 91-92: if any previous publication is online, it is suggested to cite it here. I found out a presentation published with the PROSPERO registration number (https://ebm.bmj.com/content/24/Suppl_1/A29.1).

4. Line 114: as the latest update on literature search was made almost a year ago (march 4th, 2022), it is suggested to update the search for the revision to include any newly published papers in the field.

5. Results: this section is lengthy and could be boring for readers. It is highly suggested to summarize the findings and move some of the numerous tables and figures to the appendix to make reading the results easier.

6. Lines 443-462: it is not routine to evaluate previous systematic reviews in a review study and the studies might be removed and the comparison could be moved to the discussion section of the manuscript.

7. Discussion: this section lacks the appropriate comparison with similar studies which is provided in the last part of the results section.

8. Conclusion: this section also could be briefer and only reflect the final message of the manuscript and the rest could be stated in the discussion.

Reviewer #5: Reviewer’s comments to Martiny et al

The authors have conducted a systematic review and meta-analyses of complications dou to endoscopic screening or follow-up after a positive screening test. The review is comprehensive, contains data from 134 papers and must have been extremely time-consuming to conduct. The data is of obvious importance as screening for colorectal cancer is implemented in more and more countries.

The amount of data may be the authors’s greatest challenge, also for accuracy. For example, I know that 1 patient died after TC f-up of a positive FOBT in the 2021 Randel trial, but I did not find that death in the figures.

Otherwise, I have some suggestions which may improve readability.

First of all - the figures and tables contains abbreviations that should generally be written in full. Table 3 may serve as an example that requires significant changes. Additionally – titles and figure legends should not contain abbreviations. As a rule of thumb – all tables and figures should be stand-alone parts of the manuscript. The colors of the Forest-plot should be explained in footnotes.

When quantifying adverse events, a comparator group should be included, to account for events that would have occurred without screening. In the present manuscript, no comparator was used, and accordingly, quantification of adverse events (apart from obvious endoscopic complications like GI bleeding and perforation) is susceptible to the baseline risk of those events (e.g. cardiopulmonary, death). Because the included papers are derived from different age-groups and different populations, and without a comparator, the results are very difficult to interpret. The authors should elaborate more about this limitation in the discussion.

Specific suggestions

Figures are not mentioned by occurrence in the text. Please re-assess.

Why was downgrading due to publication bias executed? What was the rationale?

4.3 belongs to methods section. In addition, how did the authors distinguish between similar subcategories, e.g. pain, discomfort, colorectal symptoms. Maybe it would be better to collapse some categories to improve readability. It should not be necessary for the reader to scrutinize the appendix to understand definitions in detail.

4.5 belongs to methods section

4.8 Occurrence of death. The authors have categorized the studies in two subpopulations depending on follow-up time was reported or not. It would make more sense if the category with f-up time reported was split into categories depending on f-up time because complications must per definition be more frequent as the f-up time increases from e.g. 7 days to 3 months.

4.8.1-4 Number of deaths per 100 000 screened should be reported. Tables should include follow-up time in categories.

4.9 What is NDCPE? Please write in full first time.

4.10 Belongs to discussion section

Consequences of CPE: This section is a pure description of the studies that reported these events, but no results. Either results should be reported, or this section should be omitted.

4.10 belongs to discussion

5. Discussion

The authors claim that severity of harm was included in their review. This is not clear to me after reading tables in section 4.9. Please clarify.

Strengths and limitation: In addition to the point I addressed previously, the authors should also mention no age categories as reasons for heterogeneity. CPE are clearly more prevalent in 75-years old compared to 45 year old individuals.

The authors should also include that most studies does not include complications from surgery due to screening findings. Especially, surgery for benign polyps that would never have caused any harm to the screenee (and could have been removed endoscopically in specialized centres) is problematic, and mortality rates of about 1%, and serious complication rate of about 15% has been reported, and may tip the harm-benefit balance in the wrong direction. Of interest, the Funen gFOBT study included death due to surgery in their analyses with the consequence that the CRC mortality reduction from screening disappeared.

Reviewer #6: In this study, which is a very difficult systematic review to understand and read, the authors aimed to review the evidence on physical  harms associated with colorectal cancer screening programs and, when possible, quantify the risk of the most serious types of physical harm during colorectal cancer screening programs i.e., deaths and cardiopulmonary events.

The authors state that they classify the study's results and do not present them all in this article. Generating Multiple Publications from One Research Study in Academic Writing: An Ethical Problem or not. Although this is also a topic of discussion, it should be mentioned in the study's limitations section. Besides, my own opinion is that it is very difficult to understand a study that includes so much data at once. Despite the fact that this circumstance is mentioned in the article, it must also be mentioned in the limitation section.

While the physical injuries are classified, they are divided into 17 subgroups, but there is considerable overlap between the groups: perforation, bleeding, colorectal complications and postpolypectomy syndrome are examples of classifications that overlap.

In addition, complications related to bowel preparation are intertwined with numerous subgroups.

This method of classification is extremely confusing.

Reviewer #7: Dear Author,

I would like to start by thanking you for sharing interesting information through this meta-analysis on physical harms associated with CCS. However, I have some comments and queries that readers may have while perusing this article. Please feel free to accept or reject the comments with rationale. The volume of comments may seem too much, but many of the comments are focused on formatting issues and clarity also.

General: I understand the hard work and effort put into extracting and presenting all the main findings identified during your review. Most authors would not have reported results with all the breakdowns and sub-categories. I appreciate the effort. However, if possible, I request that you go through your manuscript and streamline or make it concise, clear, and present it systematically. The following comments might also clarify what changes or updates can be made.

1) Keywords: There are too many keywords right now. Limit the number of keywords to around 8-10. Remove "Grade" and "Robins" keywords. The PubMed Mesh database can be used to generate proper keywords if necessary.

2) Remove the "design" subheading in the abstract and move the sentence under it to the "Methods" heading.

3) On the title page, please write down the city and country too in the affiliations where missing. Otherwise, in the future, those missed out will only trigger one more round of review/revisions. Fit the title page on one page, if possible.

4) Section 3.2: a) If there were no differences (other than dates) in the two searches (original and updated search), I suggest ignoring reporting two searches. Instead, authors may report that the search was done from inception to March 2022 (Suggestion). b) In this paragraph, clarify whether grey literature, trial registries, and references of identified papers were searched or not, and explicitly mention what type of study design was included here.

5) These days, meta-analysis papers are evaluated using the Amstar-2 checklist (Refer: https://amstar.ca/Amstar_Checklist.php) to consider them as high quality or not, in order to include them as high-quality evidence for policy recommendation, etc. Therefore, if possible, I suggest that you go through the 16 criteria and address any missing criteria in your manuscript. To be specific, item 10: reporting on the sources of funding for the studies selected in your review, 14: explanation and investigation of sources of any heterogeneity in the results, 15: detail of publication bias assessment seems to be missing. If addressed or provided rational reasons, this can be considered high-quality work by future researchers.

6) Line 130: Authors mention "In case we could not retrieve full-text articles, we included the abstracts." Please clarify how MA was done with the abstract only or remove this sentence if incorrect.

7) Please note that all appendices are included in S1 (supplementary information file 1). Therefore, please change your citation to inform that it is inside S1. For instance, "are listed in Appendix 3" to be written as "are listed in S1 Appendix 3". The same goes for figures that will not be in the main manuscript. For example, Fig 33 should be written as S2 Fig 33.

8) Authors have around 33 figures. Only a few of these figures are planned to be in the main manuscript. Other figures will be in a supplementary info file (suppose S2 List of additional figures). S2 will be a PDF consisting of a table of content (with a list of figures) and shall consist of all additional figures in it. This additional S2 file will not be formatted by the journal and will be kept as it is. Therefore, all figures inside the S2 file shall have captions, legends, and footnotes together. Do not submit additional figures as each individual figure (in TIFF format). For the figures to be in the main manuscript, it shall be submitted individually in TIFF format, while its captions and footnotes shall be in the manuscript itself.

9) Line 160-163: Avoid using bullets or lists in the main manuscript as much as possible and convert those to paragraph format. For example, it could be written as – “…excluding the following items: 1 baseline risk; 2 relative risk; 3 absolute risk; and 4 study design.”. Similar for line 258 to 273, etc.

10) Line 209: The authors mention combining RCT and NRS as they were one-armed. Please clarify for readers unfamiliar with the term “one-armed”.

11) Line 214: The authors mention calculating weights as the study size divided by the total population size across studies. Provide a rationale or reference for such usage.

12) Line 219: Author mentions “and related statistics”. Please mention what other statistics (chi-2, Q, p-value?) or remove vague terms.

13) Line 220: For calculation of CI using the Clopper-Pearson method, provide references.

14) Line 223: “These are illustrated via the orange and blue lines in the Forrest plot”. Please note that figures and table shall be stand alone, therefore, add such info in the footnotes of all relevant figure itself. Footnotes for fig in main manuscript shall be below the fig caption in the main text or footnotes can be in the figure itself.

15) The authors did not provide information on publication bias assessment in the main text method section. Please include the rationale for not conducting the assessment and why a -1 scoring was given for all.

16) There is a discrepancy between line 277, which states that 57 of the 151 subpopulations had death assessment, and the third row of table 1, which shows 39 subpopulations. Please clarify or correct this inconsistency.

17) Table 1 (also applicable to most of the tables and figures): Overall the quality, information, labels, footnotes etc. of tables and figures needs major attention and improvements. Tables and figures shall be stand alone. All the abbreviation used should be clarified in footnotes of table too (although clarified in the manuscript already). The captions of the figures and tables should be detailed enough to understand what is being presented without having to review the main text. Replace "outcome" with "death" in table 1 as it is for death assessment. Clarify that the values presented in the second and third rows of data are N and %, respectively. Clarify the difference in data presented in the "subpop with outcome assessed" and "outcome assessor" rows. Where is the asterisk symbol in the table1 data itself? Probably, no need to use boldface for all row heading and footnotes.

18) Table 3 should provide the full form of abbreviations such as AFU, FUR, NRFU, FS, etc. in the footnotes.

19) Table 3 and Table 4 can be combined into one as there are already too many tables and figures in the main manuscript.

20) Line 324's sentence is not grammatically clear. Please check and revise it. “We downgraded the

evidence with -2 due to serious risk of bias in > 50% of studies for 25 analyses (89%) and ….”.

21) In Table 5, it would be helpful to include the number of subpopulations and sample size for each pooled estimate. For instance: “6 [1-45] (28%) 9 Subpop, 2123 people”. The meanings of "critical" and "non-critical" should be explained. The full form of missing abbreviations should be included in the footnotes. Instead of having "N per 100000 [95% CI] (I2)" in the title, it can be written in the first cell and clarified in the footnotes. Additionally, if available, GRADE for each estimate can be shown using symbols such as * and ** (for low grade, medium grade etc) to provide readers with information on the pooled estimate and its certainty.

22) The captions for figures 3-8 should be revised to be more detailed and clearer as standalone explanations. For example, "Fig 3 Pooled estimate and forest plot of deaths with any follow-up time associated with sigmoidoscopy only." In the footnotes, explain that blue represents xxx while red represents xxx and clarify what "critical" means.

23) Caption of fig 5 should be moved to line 360, and adjustments should be made to other figures' caption placements accordingly.

24) The authors mention that follow-up time is categorized as short-term for less than 14 days and long-term for 0 to 30 days on line 387. Please clarify why there is overlap.

25) Table 6 needs the same improvements as Table 5, and the full form of ACS and TE should be included in the footnotes.

26) The section "4.9.1 Sigmoidoscopy" should be detailed as "4.9.1 Sigmoidoscopy and CPEs" to avoid confusion and provide clarity to readers skimming through the manuscript.. Similar approaches for this kind of section title.

27) Table 7 and 8 repeats the information provided in table 6 and respective forest plots, therefore, these two tables can be safely dropped.

28) Line 476 mentions that estimates range from 0 to 134 across meta-analyses. Please clarify whether this is across meta-analyses or across studies, as this range is not seen in result Table 5.

29) The supplementary file does not include a list of the 134 studies with brief details that meet Amstar criteria. Please provide this information.

30) To avoid potential concerns from publication teams, please convert the table in the acknowledgement section to a paragraph.

31) Check references and add missing information for e.g. for 1(missing publisher, organization), 39, 45, 50 (delete text in line 749) , 59, 61 (missing journal name) etc.

32) Please delete the supplementary file "S2 General information about the systematic review" as the information is already included in the main manuscript or submission system i.e., funding, data availability, competing interest etc. Additionally, please remove the file "changes from 2021 to 2022 version" from the next submission

33) The format of listing supplementary files in section 9 is unclear and too lengthy. Please consider listing them as follows: [S1 Appendices (includes all appendices from 1 to 22); S2 Additional Supplementary Figures (contains Fig 14 to Fig 33); S3 Prospero Protocol; S4 Prisma-harms Checklist].

34) The contents of section 10 (lines 862 to 917) are unnecessary and should be deleted as it provides a table of content for the main manuscript.

35) Once again, Please review the format of the supplementary file carefully as it will be published as is. Especially font size, style, headings size and style, heading numbering etc are all not uniform.

36) S1 is the main supplementary file and should be a bookmarked PDF. To achieve this, simply save the file as a PDF with bookmarks in Word.

37) In S1, please address the following issues: 1) Resolve the two errors on page 2. 2) Include Fig 2 on page 3 directly in the PDF file rather than submitting it separately. 3) For Appendix 3 on page 31, rather than sharing the list of excluded papers in a Google Drive link, consider including the list in the PDF or using a permanent link such as the "figshare" repository. 4) Please present numerical values in decimal format as standard practice, rather than using a comma (e.g., 0.842 instead of 0,842).

6. PLOS authors have the option to publish the peer review history of their article (what does this mean?). If published, this will include your full peer review and any attached files.

Reviewer #1: **Yes: **Ricardo Ney Cobucci

Reviewer #2: No

Reviewer #3: No

Reviewer #4: **Yes: **Sina Azadnajafabad, MD, MPH

Reviewer #5: No

Reviewer #6: **Yes: **Cihangir Akyol, MD

Reviewer #7: No

---

## [Author Response · Author response to Decision Letter 0]

9 Aug 2023

Dear Ms. Divya Bhandari,

Thank you for considering our manuscript for publication in PLOS One. In the attached files to this re-submission, we respond to your comments about the journal requirement and to the comments from the peer reviewers. We sincerely appreciate the thoughtful comments and valuable questions regarding our work from the reviewers. The feedback has been very helpful in guiding our revisions and addressing any areas that required clarification. We are grateful for the reviewer’s recognition of the hard work and effort we have invested in extracting and presenting the main findings of our review. We hope that you will find that we have taken the comments and questions raised by you and the reviewers adequately into account and that we have made the necessary changes to ensure a more coherent and reader-friendly article.

Best wishes, Frederik Martiny

---

## [Decision Letter · Decision Letter 1]

12 Nov 2023

PONE-D-22-34235R1Physical harms associated with sigmoidoscopy and colonoscopy during colorectal cancer screening - a systematic review with meta-analyses of deaths and cardiopulmonary eventsPLOS ONE

Dear Dr. Martiny,

Thank you for submitting your manuscript to PLOS ONE. After careful consideration, we feel that it has merit but does not fully meet PLOS ONE’s publication criteria as it currently stands. Therefore, we invite you to submit a revised version of the manuscript that addresses the points raised during the review process.

We look forward to receiving your revised manuscript.

Kind regards,

Divya Bhandari

Academic Editor

PLOS ONE

Journal Requirements:

Additional Editor Comments (if provided):

Thank you for the thorough revision of the manuscript, incorporating all the feedback from the reviewers. Having carefully examined the comments, responses, and revisions, I have identified a few remaining minor issues that require attention. Please note that I served as a reviewer too for this manuscript, and you can locate my comments under Reviewer 7. I anticipate that addressing these minor revisions will not take much time, and the subsequent final review stage should proceed more expeditiously. Addressing any typos, grammatical errors, or language improvements now could save time and effort in the subsequent stages.

Reviewers' comments:

Reviewer's Responses to Questions

**Comments to the Author**

1. If the authors have adequately addressed your comments raised in a previous round of review and you feel that this manuscript is now acceptable for publication, you may indicate that here to bypass the “Comments to the Author” section, enter your conflict of interest statement in the “Confidential to Editor” section, and submit your "Accept" recommendation.

Reviewer #1: All comments have been addressed

Reviewer #3: All comments have been addressed

Reviewer #4: All comments have been addressed

Reviewer #5: All comments have been addressed

Reviewer #6: All comments have been addressed

Reviewer #7: All comments have been addressed

2. Is the manuscript technically sound, and do the data support the conclusions?

Reviewer #1: Yes

Reviewer #3: Partly

Reviewer #4: Yes

Reviewer #5: (No Response)

Reviewer #6: Yes

Reviewer #7: Yes

3. Has the statistical analysis been performed appropriately and rigorously? 

Reviewer #1: Yes

Reviewer #3: No

Reviewer #4: Yes

Reviewer #5: (No Response)

Reviewer #6: Yes

Reviewer #7: Yes

4. Have the authors made all data underlying the findings in their manuscript fully available?

Reviewer #1: Yes

Reviewer #3: No

Reviewer #4: Yes

Reviewer #5: (No Response)

Reviewer #6: Yes

Reviewer #7: Yes

5. Is the manuscript presented in an intelligible fashion and written in standard English?

Reviewer #1: Yes

Reviewer #3: No

Reviewer #4: Yes

Reviewer #5: (No Response)

Reviewer #6: Yes

Reviewer #7: Yes

6. Review Comments to the Author

Reviewer #1: The authors complied with most of the recommendations made by the reviewers and justified with good arguments when they decided to keep the original text. Congratulations, as the manuscript is better for publication.

Reviewer #3: Dear authors,you have failed to present;

1. Single standing title.

2. Establishing strong niche

3. Flawed methods for the eligibility criteria, language description, consistency of tools, third reviewer, self selecting as 'author',not ise of terms,how the studies selected...etc.

54. Wide and non objective based lacking logical flow of results

5.The discussion and conclusion are also less strong to the title

Regards,

Reviewer #4: With many thanks for the efforts made to revise the manuscript based on the comments and the responses to the raised queries, I have no further comments and suggestions and I endorse publication of this submission.

Reviewer #5: (No Response)

Reviewer #6: (No Response)

Reviewer #7: Thank you for addressing all of my 37 comments provided in the earlier version.

Thanks for updating keywords with Mesh as per my comment C1. However, please remove any unnecessary sub-headings and sub-tags followed by “/”, so that the final version of keywords appears as follows:

Colorectal Neoplasms, Early Detection of Cancer, Risk, Sigmoidoscopy, Colonoscopy, Mass screening, Meta-Analysis, Systematic Review

The footnotes in the tables are currently formatted in boldface. It may be advisable to consider removing the bold formatting for improved readability.

I acknowledge that, based on reviewer's comment, the authors have suggested a more concise title, namely i.e., “Deaths and cardiovascular events following colorectal cancer screening – a systematic review with meta-analyses”. However, I have a reservation about the term "cardiovascular" not aligning entirely with the content of the manuscript, as the authors have exclusively utilized the term "cardiopulmonary" within the text. I kindly request that you revisit this aspect. I am open to retaining the original title or any other title that authors believe accurately represents your research. Therefore, please give due attention to the manuscript's title during the final technical review or proofreading stages.

7. PLOS authors have the option to publish the peer review history of their article (what does this mean?). If published, this will include your full peer review and any attached files.

Reviewer #1: **Yes: **Ricardo Ney Cobucci

Reviewer #3: No

Reviewer #4: **Yes: **Sina Azadnajafabad, MD, MPH

Reviewer #5: No

Reviewer #6: **Yes: **Cihangir Akyol, MD

Reviewer #7: **Yes: **Divya Bhandari

---

## [Author Response · Author response to Decision Letter 1]

25 Nov 2023

Dear Ms. Divya Bhandari,

Thank you for considering our responses to peer review and our revised manuscript for publication in PLOS One. Below, we respond to your comments about the journal requirement and to the comments from the peer reviewers. We have replied to each of the peer reviewers’ comments in bold font. Additionally, we have looked through the manuscript and the additional files for any remaining typos or grammatical errors, and we have improved the language throughout. 

Apologies for not having explicitly noted that you served as reviewer #7. We were conscious of this, but now I see that we did not explicitly address this. Thank you for your helpful comments.

We have corrected minor spelling mistakes and re-read the entire manuscript, having made minor changes of wording to improve the language. Also, we have re-formatted the reference to “CONSORT Harms 2022 statement, explanation, and elaboration: updated guideline for the reporting of harms in randomised trials” by Daniela and colleagues. All changes are marked in tracked changes.

RESPONSE TO REVIEWERS

Reviewer #1

The authors complied with most of the recommendations made by the reviewers and justified with good arguments when they decided to keep the original text. Congratulations, as the manuscript is better for publication.

Response:

Thank you for acknowledging our efforts to address your recommendations and for recognizing the improvements made to the manuscript. We appreciate your congratulations and are grateful for your positive assessment.

Reviewer #3

Dear authors,you have failed to present;

1. Single standing title.

2. Establishing strong niche

3. Flawed methods for the eligibility criteria, language description, consistency of tools, third reviewer, self selecting as 'author',not ise of terms,how the studies selected...etc.

54. Wide and non objective based lacking logical flow of results

5.The discussion and conclusion are also less strong to the title

Regards,

Response:

Dear colleague, we believe we have made major changes to the manuscript according corrections and recommendations from all peer reviewers that are not recognized here. In some cases, we have not been able to fulfil all suggestions. In such cases, we have explained our reasons for keeping the original wording as recognized reviewer#1. Concerning the points raised here, we would like to highlight the following:

1) To comply with the previous suggestion and the current comment by reviewer #7, we have now changed the title to: “Deaths and cardiopulmonary events following colorectal cancer screening – a systematic review with meta-analyses”. We believe that this is a single standing title in line with the title of similar reviews that adequately describes the main goal of the review.

2) We have revised the introduction in the abstract to clarify the niche that we address: “Therefore, we aimed to review the evidence on physical harms associated with endoscopic diagnostic procedures during CRCSPs and, when possible, to quantify the risk of the most serious types of physical harm during CRCSPs, i.e. deaths and cardiopulmonary events (CPEs).” In addition, the introduction establishes the rationale for the review from lines 78-94.

3) The methods section follows recommendations from the Cochrane Handbook [34], the PRISMA-harms extension and the PRISMA 2020 reporting guideline [30, 35], the AMSTAR checklist [36] and scientific literature concerning the methodological challenges of reviewing the harms of interventions [23]. We would like to highlight that we have addressed all comments regarding methods from reviewer 3 in the last round of peer review. These new points raised here are difficult to address because they are unspecific and without examples and explanations. Shortly, we do not find that the eligibility criteria are flawed, the language has been revised throughout the manuscript and is quite standard for reviews, and we consistently use the same tools in the review, which are also used in similar reviews. It is unclear to us what is meant with the following: “third reviewer, self selecting as 'author',not ise of terms,how the studies selected...etc.”

4) The results section was substantially revised following the last round of peer review with revision of tables and wording throughout following helpful suggestions from many of the reviewers. However, it is a difficult task to present the results in a simple manner due to the poor quality and the heterogeneity of the evidence as noted in our response to a similar comment form the peer reviewer in the last round of peer review: “We agree. We have followed the best-available guidelines for conduct and reporting in the area of harms of medical interventions. We acknowledge that the results section is lengthy. However, the evidence in this area is complex, heterogeneous and in general of poor quality, which has made it difficult to present it in a denser format. If the reviewer has any specific recommendations for improving clarity, we welcome these.”

5) We would like to refer to our earlier response concerning the conclusion and the discussion, which has also been revised according to recommendations from other peer reviewers: “We agree. We have provided a summary of findings that corresponds to the seven aims of the review and it is based on our results section. We go on to discuss the strengths and limitations of the evidence in the area and of our methods, followed by a comparison to similar types of studies, which is followed by recommendations for practice and future research, which we find is based on our findings. We conclude with our review questions. Again, if the reviewer has any specific recommendations for improving clarity, we welcome these.”

Reviewer #4

With many thanks for the efforts made to revise the manuscript based on the comments and the responses to the raised queries, I have no further comments and suggestions and I endorse publication of this submission.

Response: Thank you very much for endorsing the publication of the manuscript and your acknowledgment of our efforts to address comments and respond to queries.

Reviewer #5: (No Response)

Reviewer #6: (No Response)

Reviewer #7

Thank you for addressing all of my 37 comments provided in the earlier version.

Response: Thank you for the very helpful comments, we find that the manuscript has been significantly improved due to your suggestions. 

Thanks for updating keywords with Mesh as per my comment C1. However, please remove any unnecessary sub-headings and sub-tags followed by “/”, so that the final version of keywords appears as follows:

Colorectal Neoplasms, Early Detection of Cancer, Risk, Sigmoidoscopy, Colonoscopy, Mass screening, Meta-Analysis, Systematic Review

Response: Thank you. We have changed the MeSH terms according to your suggestion. 

The footnotes in the tables are currently formatted in boldface. It may be advisable to consider removing the bold formatting for improved readability.

Response: Thank you for this suggestion. We have reformatted so it is without bold face. 

I acknowledge that, based on reviewer's comment, the authors have suggested a more concise title, namely i.e., “Deaths and cardiovascular events following colorectal cancer screening – a systematic review with meta-analyses”. However, I have a reservation about the term "cardiovascular" not aligning entirely with the content of the manuscript, as the authors have exclusively utilized the term "cardiopulmonary" within the text. I kindly request that you revisit this aspect. I am open to retaining the original title or any other title that authors believe accurately represents your research. Therefore, please give due attention to the manuscript's title during the final technical review or proofreading stages.

Response: Thank you for noting that we wrote cardiovascular and not cardiopulmonary as used in the main text. We have revised the title to: “Deaths and cardiopulmonary events following colorectal cancer screening – a systematic review with meta-analyses”.

---

## [Editor Report · Decision Letter 2]

4 Dec 2023

Deaths and Cardiopulmonary Events Following Colorectal Cancer Screening – a Systematic Review with Meta-analyses

PONE-D-22-34235R2

Dear Dr. Martiny,

We’re pleased to inform you that your manuscript has been judged scientifically suitable for publication and will be formally accepted for publication once it meets all outstanding technical requirements.

Kind regards,

Divya Bhandari,

Academic Editor

PLOS ONE
---

## [Editor Report · Acceptance letter]

25 Jan 2024

PONE-D-22-34235R2 

PLOS ONE

Dear Dr. Martiny, 

I'm pleased to inform you that your manuscript has been deemed suitable for publication in PLOS ONE. Congratulations! Your manuscript is now being handed over to our production team.

Kind regards, 

on behalf of

Ms. Divya Bhandari 

Academic Editor

PLOS ONE